# Combining Generative and Discriminative Models for Hybrid Inference

**Victor Garcia Satorras**
UvA-Bosch Delta Lab
University of Amsterdam
Netherlands
v.garciasatorras@uva.nl

**Zeynep Akata** *
Cluster of Excellence ML
University of Tübingen
Germany
zeynep.akata@uni-tuebingen.de

**Max Welling**
UvA-Bosch Delta Lab
University of Amsterdam
Netherlands
m.welling@uva.nl

## Abstract

A graphical model is a structured representation of the data generating process. The traditional method to reason over random variables is to perform inference in this graphical model. However, in many cases the generating process is only a poor approximation of the much more complex true data generating process, leading to suboptimal estimations. The subtleties of the generative process are however captured in the data itself and we can "learn to infer", that is, learn a direct mapping from observations to explanatory latent variables. In this work we propose a hybrid model that combines graphical inference with a learned inverse model, which we structure as in a graph neural network, while the iterative algorithm as a whole is formulated as a recurrent neural network. By using cross-validation we can automatically balance the amount of work performed by graphical inference versus learned inference. We apply our ideas to the Kalman filter, a Gaussian hidden Markov model for time sequences, and show, among other things, that our model can estimate the trajectory of a noisy chaotic Lorenz Attractor much more accurately than either the learned or graphical inference run in isolation.

## 1   Introduction

Before deep learning, one of the dominant paradigms in machine learning was graphical models [4, 27, 21]. Graphical models structure the space of (random) variables by organizing them into a dependency graph. For instance, some variables are parents/children (directed models) or neighbors (undirected models) of other variables. These dependencies are encoded by conditional probabilities (directed models) or potentials (undirected models). While these interactions can have learnable parameters, the structure of the graph imposes a strong inductive bias onto the model. Reasoning in graphical models is performed by a process called probabilistic inference where the posterior distribution, or the most probable state of a set of variables, is computed given observations of other variables. Many approximate algorithms have been proposed to solve this problem efficiently, among which are MCMC sampling [29, 33], variational inference [18] and belief propagation algorithms [10, 21].

Graphical models are a kind of generative model where we specify important aspects of the generative process. They excel in the low data regime because we maximally utilize expert knowledge (a.k.a. inductive bias). However, human imagination often falls short of modeling all of the intricate details of the true underlying generative process. In the large data regime there is an alternative strategy which we could call "learning to infer". Here, we create lots of data pairs $\{x_n, y_n\}$ with $\{y_n\}$ the observed variables and $\{x_n\}$ the latent unobserved random variables. These can be generated from the generative model or are available directly in the dataset. Our task is now to learn a flexible

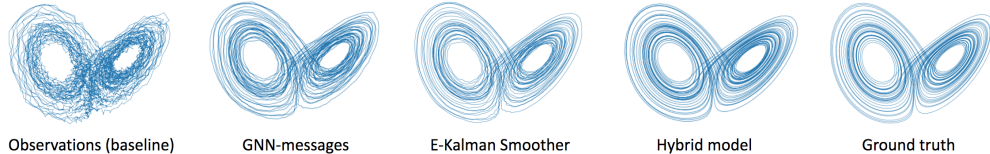

| Observations (baseline) | GNN-messages | E-Kalman Smoother | Hybrid model | Ground truth |

Figure 1: Examples of inferred 5K length trajectories for the Lorenz attractor with $\Delta t = 0.01$ trained on 50K length trajectory. The mean squared errors from left to right are (Observations: 0.2462, GNN: 0.0613, E-Kalman Smoother: 0.0372, Hybrid: 0.0169).

mapping $q(x|y)$ to infer the latent variables directly from the observations. This idea is known as "inverse modeling" in some communities. It is also known as "amortized" inference [32] or recognition networks in the world of variational autoencoders [18] and Helmholtz machines [11].

In this paper we consider inference as an iterative message passing scheme over the edges of the graphical model. We know that (approximate) inference in graphical models can be formulated as message passing, known as belief propagation, so this is a reasonable way to structure our computations. When we unroll these messages for $N$ steps we have effectively created a recurrent neural network as our computation graph. We will enrich the traditional messages with a learnable component that has the function to correct the original messages when there is enough data available. In this way we create a hybrid message passing scheme with prior components from the graphical model and learned messages from data. The learned messages may be interpreted as a kind of graph convolutional neural network [5, 15, 20].

Our Hybrid model neatly trades off the benefit of using inductive bias in the small data regime and the benefit of a much more flexible and learnable inference network when sufficient data is available. In this paper we restrict ourselves to a sequential model known as a hidden Markov process.

## 2 The Hidden Markov Process

In this section we briefly explain the Hidden Markov Process and how we intend to extend it. In a Hidden Markov Model (HMM), a set of unobserved variables $\mathbf{x} = \{x_1, \ldots, x_K\}$ define the state of a process at every time step $0 < k < K$. The set of observable variables from which we want to infer the process states are denoted by $\mathbf{y} = \{y_1, \ldots y_K\}$. HMMs are used in diverse applications as localization, tracking, weather forecasting and computational finance among others. (in fact, the Kalman filter was used to land the eagle on the moon.)

We can express $p(\mathbf{x}|\mathbf{y})$ as the probability distribution of the hidden states given the observations. Our goal is to find which states $\mathbf{x}$ maximize this probability distribution. More formally:

$$\hat{\mathbf{x}} = \arg\max_{\mathbf{x}} p(\mathbf{x}|\mathbf{y}) \tag{1}$$

Under the Markov assumption i) the transition model is described by the transition probability $p(x_t|x_{t-1})$, and ii) the measurement model is described by $p(y_t|x_t)$. Both distributions are stationary for all $k$. The resulting graphical model can be expressed with the following equation:

$$p(\mathbf{x}, \mathbf{y}) = p(x_0) \prod_{k=1}^{K} p(x_k|x_{k-1})p(y_k|x_k) \tag{2}$$

One of the best known approaches for inference problems in this graphical model is the Kalman Filter [17] and Smoother [31]. The Kalman Filter assumes both the transition and measurement distributions are linear and Gaussian. The prior knowledge we have about the process is encoded in linear transition and measurement processes, and the uncertainty of the predictions with respect to the real system is modeled by Gaussian noise:

$$x_k = \mathbf{F}x_{k-1} + q_k \tag{3}$$
$$y_k = \mathbf{H}x_k + r_k \tag{4}$$

Here $q_k, r_k$ come from Gaussian distributions $q_k \sim \mathcal{N}(0, \mathbf{Q})$, $r_k \sim \mathcal{N}(0, \mathbf{R})$. $\mathbf{F}$, $\mathbf{H}$ are the linear transition and measurement functions respectively. If the process from which we are inferring $\mathbf{x}$ is

actually Gaussian and linear, a Kalman Filter + Smoother with the right parameters is able to infer the optimal state estimates.

The real world is usually non-linear and complex, assuming that a process is linear may be a strong limitation. Some alternatives like the Extended Kalman Filter [24] and the Unscented Kalman Filter [34] are used for non-linear estimation, but even when functions are non-linear, they are still constrained to our knowledge about the dynamics of the process which may differ from real world behavior.

To model the complexities of the real world we intend to learn them from data through flexible models such as neural networks. In this work we present an hybrid inference algorithm that combines the knowledge from a generative model (e.g. physics equations) with a function that is automatically learned from data using a neural network. In our experiments we show that this hybrid method outperforms the graphical inference methods and also the neural network methods for low and high data regimes respectively. In other words, our method benefits from the inductive bias in the limit of small data and also the high capacity of a neural networks in the limit of large data. The model is shown to gracefully interpolate between these regimes.

## 3    Related Work

The proposed method has interesting relations with meta learning [2] since it learns more flexible messages on top of an existing algorithm. It is also related to structured prediction energy networks [3] which are discriminative models that exploit the structure of the output. Structured inference in relational outputs has been effective in a variety of tasks like pose estimation [35], activity recognition [12] or image classification [28]. One of the closest works is Recurrent Inference Machines (RIM) [30] where a generative model is also embedded into a Recurrent Neural Network (RNN). However in that work graphical models played no role. In the same line of learned recurrent inference, our optimization procedure shares similarities with Iterative Amortized Inference [25], although in our work we are refining the gradient using a hybrid setting while they are learning it.

Another related line of research is the convergence of graphical models with neural networks, [26] replaced the joint probabilities with trainable factors for time series data. Learning the messages in conditional random fields has been effective in segmentation tasks [7, 37]. Relatedly, [16] runs message passing algorithms on top of a latent representation learned by a deep neural network. More recently [36] showed the efficacy of using Graph Neural Networks (GNNs) for inference on a variety of graphical models, and compared the performance with classical inference algorithms. This last work is in a similar vein as ours, but in our case, learned messages are used to correct the messages from graphical inference. In the experiments we will show that this hybrid approach really improves over running GNNs in isolation.

The Kalman Filter is a widely used algorithm for inference in Hidden Markov Processes. Some works have explored the direction of coupling them with machine learning techniques. A method to discriminatively learn the noise parameters of a Kalman Filter was introduced by [1]. In order to input more complex variables, [14] back-propagates through the Kalman Filter such that an encoder can be trained at its input. Similarly, [9] replaces the dynamics defined in the Kalman Filter with a neural network. In our hybrid model, instead of replacing the already considered dynamics, we simultaneously train a learnable function for the purpose of inference.

## 4    Model

We cast our inference model as a message passing scheme where the nodes of a probabilistic graphical model can send messages to each other to infer estimates of the states $\mathbf{x}$. Our aim is to develop a hybrid scheme where messages derived from the generative graphical model are combined with GNN messages:

**Graphical Model Messages (GM-messages):** These messages are derived from the generative graphical model (e.g. equations of motion from a physics model).

**Graph Neural Network Messages (GNN-messages):** These messages are learned by a GNN which is trained to reduce the inference error on labelled data in combination with the GM-messages.

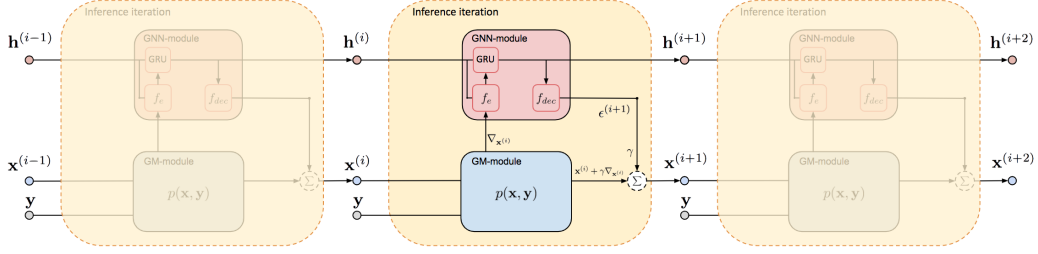

Figure 2: Graphical illustration of our Hybrid algorithm. The GM-module (blue box) sends messages to the GNN-module (red box) which refines the estimation of $\mathbf{x}$.

In the following two subsections we introduce the two types of messages and the final hybrid inference scheme.

## 4.1 Graphical Model Messages

In order to define the GM-messages, we interpret inference as an iterative optimization process to estimate the maximum likelihood values of the states $\mathbf{x}$. In its more generic form, the recursive update for each consecutive estimate of $\mathbf{x}$ is given by:

$$\mathbf{x}^{(i+1)} = \mathbf{x}^{(i)} + \gamma \nabla_{\mathbf{x}^{(i)}} \log(p(\mathbf{x}^{(i)}, \mathbf{y})) \tag{5}$$

Factorizing equation 5 to the hidden Markov Process from equation 2, we get three input messages for each inferred node $x_k$:

$$
\begin{aligned}
x_k^{(i+1)} &= x_k^{(i)} + \gamma \mathrm{M}_k^{(i)} \\
\mathrm{M}_k^{(i)} &= \mu_{x_{k-1} \to x_k}^{(i)} + \mu_{x_{k+1} \to x_k}^{(i)} + \mu_{y_k \to x_k}^{(i)}
\end{aligned}
\tag{6}
$$

$$\mu_{x_{k-1} \to x_k}^{(i)} = \frac{\partial}{\partial x_k^{(i)}} \log(p(x_k^{(i)} | x_{k-1}^{(i)})) \tag{7}$$

$$\mu_{x_{k+1} \to x_k}^{(i)} = \frac{\partial}{\partial x_k^{(i)}} \log(p(x_{k+1}^{(i)} | x_k^{(i)})) \tag{8}$$

$$\mu_{y_k \to x_k}^{(i)} = \frac{\partial}{\partial x_k^{(i)}} \log(p(y_k | x_k^{(i)})) \tag{9}$$

These messages can be obtained by computing the three derivatives from equations 7, 8, 9. It is often assumed that the transition and measurement distributions $p(x_k | x_{k-1})$, $p(y_k | x_k)$ are linear and Gaussian (e.g. Kalman Filter model). Next, we provide the expressions of the GM-messages when assuming the linear and Gaussian functions from equations 3, 4:

$$\mu_{x_{k-1} \to x_k} = -\mathbf{Q}^{-1}(x_k - \mathbf{F}x_{k-1}) \tag{10}$$

$$\mu_{x_{k+1} \to x_k} = \mathbf{F}^T \mathbf{Q}^{-1}(x_{k+1} - \mathbf{F}x_k) \tag{11}$$

$$\mu_{y_k \to x_k} = \mathbf{H}^T \mathbf{R}^{-1}(y_k - \mathbf{H}x_k) \tag{12}$$

## 4.2 Adding GNN-messages

We call $\mathbf{v}$ the collection of nodes of the graphical model $\mathbf{v} = \mathbf{x} \cup \mathbf{y}$. We also define an equivalent graph where the GNN operates by propagating the GNN messages. We build the following mappings from the nodes of the graphical model to the nodes of the GNN: $\mathbf{h_x} = \{\phi(x) : x \in \mathbf{x}\}$, $\mathbf{h_y} = \{\phi(y) : y \in \mathbf{y}\}$. Analogously, the union of both collections would be $\mathbf{h_v} = \mathbf{h_x} \cup \mathbf{h_y}$. Therefore, each node of the graphical model has a corresponding node $h$ in the GNN. The edges for both graphs are also equivalent. Values of $\mathbf{h_x}^{(0)}$ that correspond to unobserved variables $\mathbf{x}$ are randomly initialized. Instead, values $\mathbf{h_y}^{(0)}$ are obtained by forwarding $y_k$ through a linear layer.

Next we present the equations of the learned messages, which consist of a GNN message passing operation. Similarly to [23, 19], a GRU [8] is added to the message passing operation to make it recursive:

$$\mathrm{m}_{k,n}^{(i)} = z_{k,n} f_e(h_{x_k}^{(i)}, h_{v_n}^{(i)}, \mu_{v_n \to x_k}) \qquad \text{(message from GNN nodes to edge factor)} \qquad (13)$$

$$\mathrm{U}_k^{(i)} = \sum_{v_n \neq x_k} \mathrm{m}_{k,n}^{(i)} \qquad \text{(message from edge factors to GNN node)} \qquad (14)$$

$$h_{x_k}^{(i+1)} = \mathrm{GRU}(\mathrm{U}_k^{(i)}, h_{x_k}^{(i)}) \qquad \text{(RNN update)} \qquad (15)$$

$$\epsilon_k^{(i+1)} = f_{dec}(h_{x_k}^{(i+1)}) \qquad \text{(computation of correction factor)} \qquad (16)$$

Each GNN message is computed by the function $f_e(\cdot)$, which receives as input two hidden states from the last recurrent iteration, and their corresponding GM-message, this function is different for each type of edge (e.g. transition or measurement for the HMM). $z_{k,n}$ takes value 1 if there is an edge between $v_n$ and $x_k$, otherwise its value is 0. The sum of messages $\mathrm{U}_k^{(i)}$ is provided as input to the GRU function that updates each hidden state $h_{x_k}^{(i)}$ for each node. The GRU is composed by a single GRU cell preceded by a linear layer at its input. Finally a correction signal $\epsilon_k^{(i+1)}$ is decoded from each hidden state $h_{x_k}^{(i+1)}$ and it is added to the recursive operation 6, resulting in the final equation:

$$x_k^{(i+1)} = x_k^{(i)} + \gamma(\mathrm{M}_k^{(i)} + \epsilon_k^{(i+1)}) \qquad (17)$$

In summary, equation 17 defines our hybrid model in a simple recursive form where $x_k$ is updated through two contributions: one that relies on the probabilistic graphical model messages $\mathrm{M}_k^{(i)}$, and $\epsilon_k^{(i)}$, that is automatically learned. We note that it is important that the GNN messages model the *"residual error"* of the GM inference process, which is often simpler than modeling the full signal. A visual representation of the algorithm is shown in Figure 2.

In the experimental section of this work we apply our model to the Hidden Markov Process, however, the above mentioned GNN-messages are not constrained to this particular graphical structure. The GM-messages can also be obtained for other arbitrary graph structures by applying the recursive inference equation 5 to their respective graphical models.

### 4.3 Training procedure

In order to provide early feedback, the loss function is computed at every iteration with a weighted sum that emphasizes later iterations, $w_i = \frac{i}{N}$, more formally:

$$Loss(\Theta) = \sum_{i=1}^{N} w_i \mathcal{L}(\mathbf{gt}, \phi(\mathbf{x}^{(i)})) \qquad (18)$$

Where function $\phi(\cdot)$ extracts the part of the hidden state $\mathbf{x}$ contained in the ground truth $\mathbf{gt}$. In our experiments we use the mean square error for $\mathcal{L}(\cdot)$. The training procedure consists of three main steps. First, we initialize $x_k^{(0)}$ at the value that maximizes $p(y_k|x_k)$. For example, in a trajectory estimation problem we set the position values of $x_k$ as the observed positions $y_k$. Second, we tune the hyper-parameters of the graphical model as it would be done with a Kalman Filter, which are usually the variance of Gaussian distributions. Finally, we train the model using the above mentioned loss (equation 18).

## 5 Experiments

In this section we compare our Hybrid model with the Kalman Smoother and a recurrent GNN. We show that our Hybrid model can leverage the benefits of both methods for different data regimes. Next we define the models used in the experiments [2]:

**Kalman Smoother**: The Kalman Smoother is the widely known Kalman Filter algorithm [17] + the RTS smoothing step [31]. In experiments where the transition function is non-linear we use the

Extended Kalman Filter + smoothing step which we will call *"E-Kalman Smoother"*.

**GM-messages**: As a special case of our hybrid model we propose to remove the learned signal $\epsilon_k^{(i)}$ and base our predictions only on the graphical model messages from eq. 6.

**GNN-messages**: The GNN model is another special case of our model when all the GM-messages are removed and only GNN messages are propagated. Instead of decoding a refinement for the current $x_k^{(i)}$ estimate, we directly estimate: $x_k^{(i)} = \mathbf{H}^\top y_k + f_{dec}(h_{x_k}^{(i)})$. The resulting algorithm is equivalent to a Gated Graph Neural Network [23].

**Hybrid model**: This is our full model explained in section 4.2.

We set $\gamma = 0.005$ and use the Adam optimizer with a learning rate $10^{-3}$. The number of inference iterations used in the Hybrid model, GNN-messages and GM-messages is N=50. $f_e$ and $f_{dec}$ are a 2-layers MLPs with Leaky Relu and Relu activations respectively. The number of features in the hidden layers of the GRU, $f_e$ and $f_{dec}$ is nf=48. In trajectory estimation experiments, $y_k$ values may take any value from the real numbers $\mathbb{R}$. Shifting a trajectory to a non-previously seen position may hurt the generalization performance of the neural network. To make the problem translation invariant we modify $y_k$ before mapping it to $h_{y_k}$, we use the difference between the observed current position with the previous one and with the next one.

## 5.1 Linear dynamics

The aim of this experiment is to infer the position of every node in trajectories generated by linear and gaussian equations. The advantage of using a synthetic environment is that we know in advance the original equations the motion pattern was generated from, and by providing the right linear and gaussian equations to a Kalman Smoother we can obtain the optimal inferred estimate as a lower bound of the test loss.

Among other tasks, Kalman Filters are used to refine the noisy measurement of GPS systems. A physics model of the dynamics can be provided to the graphical model that, combined with the noisy measurements, gives a more accurate estimation of the position. The real world is usually more complex than the equations we may provide to our graphical model, leading to a gap between the assumed dynamics and the real world dynamics. Our hybrid model is able to fill this gap without the need to learn everything from scratch.

To show that, we generate synthetic trajectories $\mathcal{T} = \{\mathbf{x}, \mathbf{y}\}$. Each state $x_k \in \mathbb{R}^6$ is a 6-dimensional vector that encodes position, velocity and acceleration $(p, v, a)$ for two dimensions. Each $y_k \in \mathbb{R}^2$ is a noisy measurement of the position also for two dimensions. The transition dynamic is a non-uniform accelerated motion that also considers drag (air resistance):

$$\frac{\partial p}{\partial t} = v, \qquad\qquad \frac{\partial v}{\partial t} = a - cv, \qquad\qquad \frac{\partial a}{\partial t} = -\tau v \qquad (19)$$

Where $-cv$ represents the air resistance [13], with $c$ being a constant that depends on the properties of the fluid and the object dimensions. Finally, the variable $-\tau v$ is used to non-uniformly accelerate the object.

To generate the dataset, we sample from the Markov process of equation 2 where the transition probability distribution $p(x_{k+1}|x_k)$ and the measurement probability distribution $p(y_k|x_k)$ follow equations (3, 4). Values $\mathbf{F}, \mathbf{Q}, \mathbf{H}, \mathbf{R}$ for these distributions are described in the Appendix, in particular, $\mathbf{F}$ is analytically obtained from the above mentioned differential equations 19. We sample two different motion trajectories from 50 to 100K time steps each, one for validation and the other for training. An additional 10K time steps trajectory is sampled for testing. The sampling time step is $\Delta t = 1$.

Alternatively, the graphical model of the algorithm is limited to a uniform motion pattern $p = p_0 + vt$. Its equivalent differential equations form would be $\frac{\partial p}{\partial t} = v$. Notice that the air friction is not considered

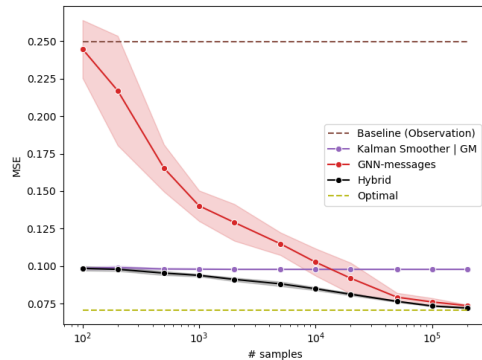

Figure 3: MSE comparison with respect to the number of training samples for the linear dynamics dataset.

anymore and velocity and acceleration are assumed to be uniform. Again the parameters for the matrices $\mathbf{F}, \mathbf{Q}, \mathbf{H}, \mathbf{R}$ when considering a uniform motion pattern are analytically obtained and described in the Appendix.

**Results.** The Mean Square Error with respect to the number of training samples is shown for different algorithms in Figure 3. The plot shows the average and the standard deviation over 7 runs, the sampled test trajectory remains the same over all runs, this is not the case for the training and validation sampled trajectories. Note that the MSE of the Kalman Smoother and GM-messages overlap in the plot since both errors were exactly the same.

Our model outperforms both the GNN or Kalman Smoother in isolation in all data regimes, and it has a significant edge over the Kalman Smoother when the number of samples is larger than 1K. This shows that our model is able to ensemble the advantages of prior knowledge and deep learning in a single framework. These results show that our hybrid model benefits from the inductive bias of the graphical model equations when data is scarce, and simultaneously it benefits from the flexibility of the GNN when data is abound.

A clear trade-off can be observed between the Kalman smoother and the GNN. The Kalman Smoother clearly performs better for low data regimes, while the GNN outperforms it for larger amounts of data (>10K). The hybrid model is able to benefit from the strengths of both.

## 5.2 Lorenz Attractor

The Lorenz equations describe a non-linear chaotic system used for atmospheric convection. Learning the dynamics of this chaotic system in a supervised way is expected to be more challenging than for linear dynamics, making it an interesting evaluation of our Hybrid model. A Lorenz system is modelled by three differential equations that define the convection rate, the horizontal temperature variation and the vertical temperature variation of a fluid:

$$\frac{\partial z_1}{\partial t} = 10(z_2 - z_1), \qquad \frac{\partial z_2}{\partial t} = z_1(28 - z_3) - z_2, \qquad \frac{\partial z_3}{\partial t} = z_1 z_2 - \frac{8}{3} z_3 \qquad (20)$$

To generate a trajectory we run the Lorenz equations 20 with a $dt = 10^{-5}$ from which we sample with a time step of $\Delta t = 0.05$ resulting in a single trajectory of 104K time steps. Each point is then perturbed with gaussian noise of standard deviation $\lambda = 0.5$. From this trajectory, 4K time steps are separated for testing, the remaining trajectory of 100K time steps is equally split between training and validation partitions.

Assuming $x \in \mathbb{R}^3$ is a 3-dimensional vector $x = [z_1, z_2, z_3]^\top$, we can write down the dynamics matrix of the system as $\mathbf{A}_{|x}$ from the Lorenz differential eq. 20, and obtain the transition function $\mathbf{F}_{|x_k}$ [22] using the Taylor Expansion.

$$\dot{x} = \mathbf{A}_{|x} x = \begin{bmatrix} -10 & 10 & 0 \\ 28 - z_3 & -1 & 0 \\ z_2 & 0 & -\frac{8}{3} \end{bmatrix} \begin{bmatrix} z_1 \\ z_2 \\ z_3 \end{bmatrix}, \quad \mathbf{F}_{|x_k} = \mathbf{I} + \sum_{j=1}^{J} \frac{(\mathbf{A}_{|x_k} \Delta t)^j}{j!} \qquad (21)$$

where $\mathbf{I}$ is the identity matrix and $J$ is the number of terms from the Taylor expansion. We run simulations for J=1, J=2 and J=5. For larger J the improvement was minimal. For the measurement model $\mathbf{H} = \mathbf{I}$ we use the identity matrix. For the noise distributions $\mathbf{Q} = \sigma^2 \Delta t \mathbf{I}$ and $\mathbf{R} = 0.5^2 \mathbf{I}$ we use diagonal matrices. The only hyper-parameter to tune from the graphical model is $\sigma$.

Since the dynamics are non-linear, the matrix $\mathbf{F}_{|x_k}$ depends on the values $x_k$. The presence of these variables inside the matrix introduces a simple non-linearity that makes the function much harder to learn.

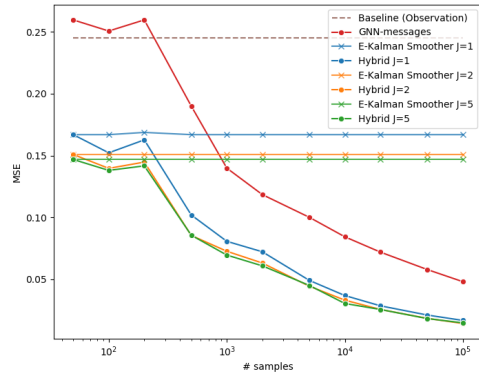

Figure 4: MSE with respect to the the number of training samples on the Lorenz Attractor.

**Results.** The results in Figure 4 show that the GNN struggles to achieve low accuracies for this chaotic

system, i.e. it does not converge together with the hybrid model even when the training dataset contains up to $10^5$ samples and the hybrid loss is already $0.01 \sim 0.02$. We attribute this difficulty to the fact the matrix $\mathbf{F}_{|x_k}$ is different at every state $x_k$, becoming harder to approximate.

This behavior is different from the previous experiment (linear dynamics) where both the Hybrid model and the GNN converged to the optimal solution for high data regimes. In this experiment, even when the GNN and the E-Kalman Smoother perform poorly, the Hybrid model gets closer to the optimal solution, outperforming both of them in isolation. This shows that the Hybrid model benefits from the labeled data even in situations where its fully-supervised variant or the E-Kalman Smoother are unable to properly model the process. One reason for this could be that the residual dynamics (i.e. the error of the E-Kalman Smoother) are much more linear than the original dynamics and hence easier to model by the GNN.

As can be seen in Figure 4, depending on the amount of prior knowledge used in our hybrid model we will need more or less samples to achieve a particular accuracy. Following, we show in table 5.2 the approximate number of training samples required to achieve accuracies 0.1 and 0.05 depending on the amount of knowledge we provide (i.e. the number of $J$ terms of the Taylor expansion). The hybrid method requires $\sim 10$ times less samples than the fully-learned method for MSE=0.1 and $\sim 20$ times less samples for MSE=0.05.

|  | GNN (J=0) | Hybrid (J=1) | Hybrid (J=2 & J=5) |
|---|---|---|---|
| MSE = 0.1 | $\sim 5.000$ | $\sim 500$ | $\sim 400$ |
| MSE = 0.05 | $\sim 90.000$ | $\sim 5.000$ | $\sim 4.000$ |

Table 1: Number of samples required to achieve a particular MSE depending on the amount of prior knowledge (i.e. J). These numbers have been extracted from Figure 4.

Qualitative results of estimated trajectories by the different algorithms on the Lorenz attractor are depicted in Figure 1. The plots correspond to a 5K length test trajectory (with $\Delta t = 0.01$). All trainable algorithms have been trained on 5K length trajectories.

## 5.3 Real World Dynamics: Michigan NCLT dataset

To demonstrate the generalizability of our Hybrid model to real world datasets, we use the Michigan NCLT [6] dataset which is collected by a segway robot moving around the University of Michigan's North Campus. It comprises different trajectories where the GPS measurements and the ground truth location of the robot are provided. Given these noisy GPS observations, our goal is to infer a more accurate position of the segway at a given time.

In our experiments we arbitrarily use the session with date 2012-01-22 which consists of a single trajectory of 6.1 Km on a cloudy day. Sampling at 1Hz results in 4.629 time steps and after removing the parts with a unstable GPS signal, 4.344 time steps remain. Finally, we split the trajectory into three sections: 1.502 time steps for training, 1.469 for validation and 1.373 for testing. The GPS measurements are assumed to be the noisy measurements denoted by $\mathbf{y}$.

| ALGORITHM | MSE |
|---|---|
| OBSERVATIONS (BASELINE) | 3.4974 |
| KALMAN SMOOTHER | 3.0099 |
| GM-MESSAGES | 3.0048 |
| GNN-MESSAGES | 1.7929 |
| HYBRID MODEL | **1.4109** |

Table 2: MSE for different methods on the Michigan NCLT datset.

For the transition and measurement graphical model distributions we assume the same uniform motion model used in section 5.1, specifically the dynamics of a uniform motion pattern. The only parameters to learn from the graphical model will be the variance from the measurement and transition distributions. The detailed equations are presented in the Appendix.

**Results.** Our results show that our Hybrid model (1.4109 MSE) outperforms the GNN (1.7929 MSE), the Kalman Smoother (3.0099 MSE) and the GM-messages (3.0048 MSE). One of the advantages of the GNN and the Hybrid methods on real world datasets is that both can model the correlations through time from the noise distributions while the GM-messages and the Kalman Smoother assume the noise to be uncorrelated through time as it is defined in the graphical model. In

summary, this experiment shows that our hybrid model can generalize with good performance to a real world dataset.

## 6    Discussion

In this work, we explored the combination of recent advances in neural networks (e.g. graph neural networks) with more traditional methods of graphical inference in hidden Markov models for time series. The result is a hybrid algorithm that benefits from the inductive bias of graphical models and from the high flexibility of neural networks. We demonstrated these benefits in three different tasks for trajectory estimation, a linear dynamics dataset, a non-linear chaotic system (Lorenz attractor) and a real world positioning system. In three experiments, the Hybrid method learns to efficiently combine graphical inference with learned inference, outperforming both when run in isolation.

Possible future directions include applying our idea to other generative models. The equations that describe our hybrid model are defined on edges and nodes, therefore, by modifying the input graph, i.e. by modifying the edges and nodes of the input graph, we can run our algorithm on arbitrary graph structures. Other future directions include exploring hybrid methods for performing probabilistic inference in other graphical models (e.g. discrete variables), as well learning the graphical model itself. In this work we used cross-validation to make sure we did not overfit the GNN part of the model to the data at hand, optimally balancing prior knowledge and data-driven inference. In the future we intend to explore a more principled Bayesian approach to this. Finally, hybrid models like the one presented on this paper can help improve the interpretability of model predictions due to their graphical model backbone.

## Footnotes

* Majority of this work has been done when Zeynep Akata was at the University of Amsterdam.

[2]Available at: https://github.com/vgsatorras/hybrid-inference

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
