[Supplementary Material]


# A  Appendix

## A.1  Equation details the for Linear dynamics experiment

**Linear and Gaussian dataset matrices:**

The differential equations that describe the dynamics are ($c = 0.06$, $\tau = 0.17$):

$$\frac{\partial p}{\partial t} = v, \qquad\qquad \frac{\partial v}{\partial t} = a - cv, \qquad\qquad \frac{\partial a}{\partial t} = -\tau v \tag{22}$$

Therefore, the dynamics matrix is defined as:

$$\dot{x} = \mathbf{A}x = \begin{bmatrix} 0 & 1 & 0 \\ 0 & -c & 1 \\ 0 & -\tau c & 0 \end{bmatrix} \begin{bmatrix} p \\ v \\ a \end{bmatrix} \tag{23}$$

Using the following Taylor expansion we find the transition matrix $\tilde{\mathbf{F}}$

$$\tilde{\mathbf{F}} = \mathbf{I} + \sum_{k=1}^{K} \frac{(\mathbf{A}\Delta t)^k}{k!} \tag{24}$$

The transition matrix $\tilde{\mathbf{F}}$ for each dimension is:

$$\tilde{\mathbf{F}} = \begin{bmatrix} 1 & \Delta t - \frac{c}{2}\Delta t^2 & \frac{\Delta t^2}{2} \\ 0 & 1 - c\Delta t + \frac{c^2 - \tau}{2}\Delta t^2 & \Delta t - \frac{c}{2}\Delta t^2 \\ 0 & -\tau c + \frac{\tau c}{2}\Delta t^2 & 1 - \frac{\tau}{2}\Delta t^2 \end{bmatrix} \tag{25}$$

Then, the transition matrix and noise distributions are:

$$\mathbf{F} = \begin{bmatrix} \tilde{\mathbf{F}} & \mathbf{0} \\ \mathbf{0} & \tilde{\mathbf{F}} \end{bmatrix}, \tilde{\mathbf{Q}} = \begin{bmatrix} \Delta t/3 & 0 & 0 \\ 0 & \Delta t & 0 \\ 0 & 0 & 3\Delta t \end{bmatrix}, \mathbf{Q} = 0.1^2 \begin{bmatrix} \tilde{\mathbf{Q}} & 0 \\ 0 & \tilde{\mathbf{Q}} \end{bmatrix},$$

The measurement matrix and noise distribution are:

$$\mathbf{H} = \begin{bmatrix} 1 & 0 & 0 & 0 & 0 & 0 \\ 0 & 0 & 0 & 1 & 0 & 0 \end{bmatrix}, \mathbf{R} = 0.5^2 \begin{bmatrix} 1 & 0 \\ 0 & 1 \end{bmatrix}$$

**Linear and Gaussian matrices used for hybrid and GM-messages models:**

The differential equations that describe the dynamics are:

$$\frac{\partial p}{\partial t} = v, \qquad\qquad \frac{\partial v}{\partial t} = 0, \qquad\qquad \frac{\partial a}{\partial t} = 0 \tag{26}$$

Then the transition matrix given the last equations is:

$$\tilde{\mathbf{F}} = \begin{bmatrix} 1 & \Delta t \\ 0 & 1 \end{bmatrix} \tag{27}$$

And the transition matrix and noise distributions are:

$$\mathbf{F} = \begin{bmatrix} \tilde{\mathbf{F}} & \mathbf{0} \\ \mathbf{0} & \tilde{\mathbf{F}} \end{bmatrix}, \tilde{\mathbf{Q}} = \begin{bmatrix} \Delta t & 0 \\ 0 & \Delta t \end{bmatrix} \mathbf{Q} = \begin{bmatrix} \sigma^2 \tilde{\mathbf{Q}} & 0 \\ 0 & \sigma^2 \tilde{\mathbf{Q}} \end{bmatrix},$$

$$\tag{28}$$

Finally the measurement distribution matrices are:

$$\mathbf{H} = \begin{bmatrix} 1 & 0 & 0 & 0 & 0 & 0 \\ 0 & 0 & 0 & 1 & 0 & 0 \end{bmatrix}, \mathbf{R} = 0.5^2 \begin{bmatrix} 1 & 0 \\ 0 & 1 \end{bmatrix}$$

Such that the only parameter to optimize from the graphical model is the variance of the transition noise distribution $\sigma$

## A.2 Equation details for the NCLT dataset

For the NCLT dataset we use the uniform velocity motion equations. The differential equations that describe the dynamics are:

$$\frac{\partial p}{\partial t} = v, \qquad\qquad \frac{\partial v}{\partial t} = 0, \qquad\qquad \frac{\partial a}{\partial t} = 0 \qquad (29)$$

Such that the transition matrix for one component is:

$$\tilde{\mathbf{F}} = \begin{bmatrix} 1 & \Delta t \\ 0 & 1 \end{bmatrix}, \qquad (30)$$

And the transition matrix and noise distribution are:

$$\mathbf{F} = \begin{bmatrix} \tilde{\mathbf{F}} & 0 \\ 0 & \tilde{\mathbf{F}} \end{bmatrix}, \tilde{\mathbf{Q}} = \begin{bmatrix} \Delta t & 0 \\ 0 & \Delta t \end{bmatrix} \mathbf{Q} = \begin{bmatrix} \sigma^2\tilde{\mathbf{Q}} & 0 \\ 0 & \sigma^2\tilde{\mathbf{Q}} \end{bmatrix},$$

$$(31)$$

Finally the measurement distribution matrices are:

$$\mathbf{H} = \begin{bmatrix} 1 & 0 & 0 & 0 \\ 0 & 0 & 1 & 0 \end{bmatrix}, \mathbf{R} = \lambda^2 \begin{bmatrix} 1 & 0 \\ 0 & 1 \end{bmatrix}$$

Such that the only parameters to optimize from the graphical model is the variance of the noise and measurement distributions $\sigma$ and $\lambda$.