[Reviews · NeurIPS 2019]

Reviewer 1



This paper is very well-written, providing a clear description of the proposed method and a compelling argument for why it would be useful. The experiments nicely demonstrate the advantages of this method (combining the advantages of generative models with those of DL for approximate inference). I really liked this paper. There is some related work in this field. As far as I know, the authors do a good job at putting this work in context and citing relevant papers. I believe this work is both original and highly significant.

Reviewer 2



Overall this is a nice idea that works on using black box models to amortize the residuals from doing inference assuming a linearized approximation to the model. I found the experiments to be well organized albeit mostly on small scale/synthetic data. Summary: This paper introduces a procedure for combining graph neural networks with traditional methods for probabilistic inference (instantiated in HMMs). When we have linear dynamics in a HMM, inference is exact. For nonlinear dynamics, when we have access to the functional form of the true dynamics of the state space model, we can linearize the transition and emission functions (via a Taylor expansion) and represent them as matrices. Using the transition and emission matrices, we can obtain the closed form update rules for the latent variables given observations. These form the “GM-messages”. However since we linearized the (potentially nonlinear) model, we may have a gap between the true posterior and the one representable by GM-messages. To fix this, the paper models the residuals from the linearized model using a graph neural network (specifically a combination of an GRU and a few MLPs). The combination of the two is used to model the updates to the latent states. The method is compared to (a) Kalman smoothing (b) a variant of the method that only uses message passing, and (c) a variant that only uses the GNN messages on three datasets: (i) a linear model where the number of samples to match the true states is tracked -- the key finding is that the hybrid model is capable of finding the optimal posterior distribution (ii) a nonlinear lorentz attractor simulation where the MSE between the true states and inferred states is tracked -- here the hybrid model realizes the lowest MSE (iii) a robotics dataset where the goal is to infer the true GPS location of the robot from ground truth measurements. Here too, the hybrid model realizes the lowest MSE. Comments: (1) One way to improve the manuscript would be to study how the proposed method scales with the dimensionality of the observations as well as how the sample complexity increases as the true transition/emission function move further away from being linear. (2) One limitation of the work that could be made more explicit is that practitioners need to know the functional form of the transition and emission function apriori in order to estimate the matrices necessary to initialize the GM-messages. (3) I think the experimental section can also be improved by adding in a comparison to Unscented Kalman Filters. Unlike EKFs which work with a linearization of nonlinear model functions, UKFs have a tunable number of particles to obtain high fidelity fits to the latent variable. It would be worthwhile incorporating this baseline into the results on the lorentz sequences as well as the nclt dataset. (4) Missing citation to related work: http://proceedings.mlr.press/v80/marino18a/marino18a.pdf

Reviewer 3



The idea of the paper is straightforward, which is to use a graph neural network as a residual part to fix the messages from a Gaussian graphical model. The proposed model is simple but effective. My main concern is the experimental part. In the experiment, the dynamics are simple and fixed. The authors may want to train the model on a set of different dynamics (with the same structure, but different parameters), and then test them on the dynamics with the same structure but different parameters (obviously, the parameters can be input to the model as conditional variables). That would make the paper more convincing. ===== After Rebuttal ===== The authors' response resolves my concern, and thus I update my score.

[Author Response · NeurIPS 2019]

We sincerely thank all the reviewers for their feedback indicating that we present an innovative work that could have a
wide impact (R1), and for appreciating the overall idea and emphasizing the effectiveness of our method (R2, R3). We
strongly believe that combining the advantages of more classical inference methods with those of Deep Learning may
be of interest to the research community and may impact future applications. We address the reviewer comments below.

**R1: More discussion about how this idea could be applied to other generative models.** We extended the paper
discussion section with a more detailed explanation of how to apply our idea to other generative models. The equations
that describe the current model of the paper are defined on edges and nodes, therefore, by modifying the input graph, in
other words, by modifying the edges and nodes of the input graph we can run the algorithm on any arbitrary structure.
Furthermore, in the future, we could experiment with different types of messages for the GM-messages, for example
Belief Propagation messages, then the model would also be able to run on discrete variables.

**R2: One way to improve the manuscript would be to study how the proposed method scales with the dimension-**
**ality of the observations as well as how the sample complexity increases as the true transition/emission function**
**move further away from being linear.** In the Lorenz attractor experiment, we are computing the Taylor expansion
from the Lorenz differential equations. The larger the number of expansion terms, the further the transition function
moves away from being linear. Currently, in the paper, we only show the plots for J=1 and J=2 expansion terms. Based
on this feedback, we will add another expansion term J=5 (for J>5 there is not noticeable improvement). And then, we
will study how the sample complexity decreases as the transition/emission function of the model moves away from
being linear.

**R2: One limitation of the work that could be made more explicit is that practitioners need to know the func-**
**tional form of the transition and emission function apriori in order to estimate the matrices necessary to initial-**
**ize the GM-messages.** A strength of our method is the capability of combining prior knowledge known by practitioners
with dynamics learned from the data. Instead of having to choose between more classical methods or deep learning,
we want to obtain the benefits from both. Therefore, we don't see as a limitation the fact that the functional form is
needed. Given the functional form, our method has the advantage to combine it with deep learning. To avoid confusions
and emphasize that GM-messages are a way of encoding the prior knowledge known by practitioners we changed
lines 113-114 from the paper to: *"Graphical Model Messages (GM-messages): These messages are derived from the*
*generative graphical model that encodes the prior knowledge known by practitioners (e.g. equations of motion from a*
*physics model)."*

**R2: I think the experimental section can also be improved by adding in a comparison to Unscented Kalman**
**Filters. Unlike EKFs which work with a linearization of nonlinear model functions, UKFs have a tunable**
**number of particles to obtain high fidelity fits to the latent variable. It would be worthwhile incorporating this**
**baseline into the results on the lorentz sequences as well as the nclt dataset.** We also thought of adding the UKF as
another baseline. But in the NCLT dataset, although the real world dynamics may be complex, the assumed dynamics
of the GM-messages are already linear, therefore, there is no need to use EKF or UKF here. In the Lorenz attractor,
there isn't a closed form function $x_{k+1} = f(x_k)$ that defines its dynamics. Instead, we are provided with the dynamics
matrix defined by the Lorenz differential equations, from which we can compute the Taylor Expansion to approximate
$x_{k+1} = \hat{f}_{x_k}(x_k)$ as explained in section 5.2 of the paper. Indeed, in case we had access to the true $f(x_k)$, by computing
its Taylor Expansion at point $x_k$ we would obtain $\hat{f}_{x_k}(x_k)$. Hence, we can apply the EKF as if we had access to
$f(x_k)$ but not the UKF. We could run UKF on an approximation of $f(x_k)$, it shouldn't be the same approximation we
used in the EKF, otherwise it would be a suboptimal solution of the EKF. We can study running the UKF in a better
approximation of $f(x_k)$ than the Taylor expansion, but maybe that diverges from the scope of our paper.

**R2: Missing citation to related work: http://proceedings.mlr.press/v80/marino18a/marino18a.pdf** We added the
paper to the related work. It has interesting relations with our paper since it also uses meta-learning to learn an
iterative inference algorithm. They repeatedly compute posterior gradients to perform optimization and they encode
this gradients with a Neural Network that learns how to update the posterior estimates. However, in our case, graphical
models play a stronger role and we combine prior knowledge with learned inference.

**R3: My main concern is the experimental part. In the experiment, the dynamics are simple and fixed. The**
**authors may want to train the model on a set of different dynamics (with the same structure, but different**
**parameters), and then test them on the dynamics with the same structure but different parameters (obviously,**
**the parameters can be input to the model as conditional variables).** Actually, the dynamics of the Lorenz attractor
depend at every time step on the current state. It means they are not fixed, instead the parameters change at every
timestep. The same happens at test time, the dynamics are different and also change at every time step depending on the
current state. Our method significantly outperformed the others in this experiment.

**R1, R2, R3:** Beside the already commented changes, we updated the experiments adding error bars to the plots and we
polished some sentences/writing.

[Meta-Review · NeurIPS 2019]

There is consensus among reviewers that this is a beautifully executed paper, with possibly a small query about experimental evaluation on larger real-world data examples. There is not much a meta-reviewer could add to the strong reviews that this paper has already received.